# Impact of Chinese Government Subsidies on Enterprise Innovation: Based on a Three-Dimensional Perspective

**Lili Jia** [1,2], **Eunyoung Nam** [3,*] and **Dongphil Chun** [2,*]

1 School of Economics and Management, Taishan University, Taian 271000, China; jll@tsu.edu.cn
2 Graduate School of Management of Technology, Pukyong National University, Busan 48547, Korea
3 College of Business Administration, Sejong University, Seoul 05006, Korea
* Correspondence: nanyinying@sejong.ac.kr (E.N.); performance@pknu.ac.kr (D.C.); Tel.: +82-51-629-5647 (D.C.)

**Abstract:** Government subsidies are an important means to guide enterprises' investment in technological innovation. While countries are increasing government subsidies to enterprises, how to effectively leverage government subsidies is a concern of the academic community. At present, scholars' research conclusions on the impact of government subsidies on enterprise technological innovation include promotion effect, extrusion effect, and mixing effect. Relevant research is often conducted from a single perspective. This paper studies the relationship between government subsidies and enterprise technological innovation, and integrates the macro-institutional environment, meso-market structure, and micro-corporate governance into the same framework. Taking information transmission, software, and information technology service companies as samples, it analyzes the influencing factors of the Chinese government research and development (R&D) subsidies on enterprises' innovation investment. This paper uses Stata16 software to perform the least square analysis. The research shows that the Chinese government R&D subsidies have a significant incentive effect on corporate technology innovation investment. The higher the marketization process, the more dispersed its equity, and the government subsidy promotes corporate technology innovation investment. The more significant it is; for industries with different product market competition, government subsidies have no significant impact on enterprises' investment in technological innovation. Based on empirical research conclusions, this study puts forward policy recommendations to increase the intensity of government subsidies and optimize the structure of corporate equity to increase the leverage effect of government subsidies.

**Keywords:** Chinese government subsidies; technology innovation investment; institutional environment; market structure; corporate governance





## 1. Introduction

Modern economic growth theory shows that technological development and knowledge accumulation play a pivotal role in determining economic growth [1]. To develop emerging industries and promote technological innovation, governments globally have formulated a series of fiscal policies. Among them, government subsidies are one of the most important methods to improve technological development. Chinese government subsidies for enterprises have increased from 18.39 billion yuan in 2009 to 49.13 billion yuan in 2018, with an average annual growth rate of 16.7%. While the scale of government subsidies is constantly increasing, Chinese enterprise's technological development is still relatively backward. Many companies still focus on low-tech, low-value-added areas [2,3]. Among companies in strategic emerging industries heavily subsidized by the government, only 9.3% are in a leading international position, less than 16% are in an advanced global position, and more than 70% cannot compete internationally. Thus, the level of technological innovation in China's strategic emerging industries is still low [4]. With natural resources, low-cost labor, and other factor endowments gradually weakening, innovation has become

a driving factor for China to build its national core competitiveness [5]. As a representative nation that has moved from a planned economy to a market economy, China is worthy of study on the impact of government subsidies on enterprise innovation. As a major source of funds for technological innovation, we investigate whether government subsidies have effectively promoted technological innovation in China. We also investigate the factors that moderate the effectiveness of Chinese government subsidies for technological innovation.

The impact of government subsidies on the enterprise's technological innovation is still controversial. Blank and Stigler investigated the relationship between government subsidies and technological innovation [6]. They found that the government can both stimulate technological innovation and replace technological innovation. However, the relationship between the two is uncertain. Many studies have investigated the effects of government subsidies on enterprises' technological innovation. Their findings are still inconclusive. Some studies believe that government subsidies have an incentive effect on technological innovation. Antonelli analyzed 86 sample data collected from Italy and found a significant positive relationship between government subsidies and enterprises' technological innovation [7]. Other scholars believe that there is a substitution effect between government subsidies and enterprises' technological innovation. Toivanen and Niininen collected data for 1989–1993 on Finnish companies and found a substitution effect in large enterprises [8]. Wallsten analyzed 81 samples in the United States and found that government subsidies did not promote enterprises' technological innovation [9].

Although arguments on whether government subsidies can stimulate technological innovation in enterprises are still inconclusive, various governments are still using subsidies to promote technological innovation. With the transition from a planned economy to a market economy, the Chinese government's industrial policy has a significant impact on firms' operating decisions. The marketization in various regions in China is extremely uneven, with different competition levels in each industry. Problems of corporate governance owing to economic and other factors are still prominent. In this case, it is particularly necessary to study the influence of government subsidies on corporate innovation.

In related studies, scholars have researched the types of enterprise. In terms of analysis from the perspective of the nature of property rights, Li Ling and Tao Houyong analyzed the data of 974 listed companies and found that government subsidies significantly impact the research and development (R&D) investment of private enterprises, and they played the role of "guiding hand". It has a positive effect, but it has no significant impact on the R&D investment of state-owned enterprises, and it plays a negative role as a "conniving hand" [9]. Some researchers are conducting from the perspective of enterprise R&D foundation. Bai Junhong uses the industrial data of China's large and medium-sized industrial enterprises from 1998 to 2007, and uses static and dynamic panel data models to study. The results found that government subsidies have a significant inducing effect on the R&D investment of enterprises. The greater the stock of knowledge and the higher the technical level, the more obvious the inducing effect [10]. From the perspective of industry competition, Lee [11] believes that the more intense the industry competition, the more likely it is for companies to obtain R&D cost reduction effects, and the greater the leverage effect of government R&D funding.

Based on existing research, this study investigates the relationship between Chinese government subsidies and technological innovation. Unlike existing research, we use the marketization index of China's provinces, product market competition, and ownership concentration to reflect the macro-institutional environment, medium-sized market structure, and micro-corporate governance, respectively. We use these indicators to establish a theoretical framework, take the information transmission, software, and information technology industry as an example. Using the least squares method, Stata 16 software (StataCorp LLC, Texas, TX, USA) was employed to analyze Chinese government subsidies on enterprise technological innovation. We also explore the theoretical mechanism behind the government subsidy's leverage.

The rest of the article is organized as follows: Section 2 reviews the previous studies that focus on the effects of subsidies on technological innovation. Section 3 describes the sample data and research design. Section 4 discusses the empirical results and analysis. Section 5 concludes the paper and provides policy recommendations.

## 2. Government Subsidies and Research and Development (R&D) Investment: A Literature Review

### 2.1. Existing Research

Corporate R&D investment is often lower than the optimal level owing to high costs and high risks associated with this investment. Thus, governments have taken charge of subsidizing R&D to induce corporate investment. The major research question has been whether government R&D subsidies are either complementary and, thus, 'additional' to company-financed R&D, or whether they substitute for and, thus, 'crowd-out' private R&D [12]. Blank and Stigler were among the first researchers to perform an empirical analysis of the relationship between publicly funded and private R&D investment. Their results were mixed, with evidence supporting both additionality and substitution effects. After almost five decades of research, the empirical evidence is mixed, and the question is far from having a conclusive answer. The disparity in results can be attributed to differences in the populations under study (periods, countries of interest, business sectors), the variables used, and the empirical approach [12–16]. Most of the relevant empirical studies were performed during the 2000s. With the increasing availability of appropriate datasets, this is a clear sign of the growing concern about the role that public subsidies play in private R&D decisions [17]. This section summarizes the effects of subsidies collected from previous studies.

#### 2.1.1. Crowding-In Effect

Promotion effects have been examined from different angles. From the perspective of externality theory, Lee [11] states that government subsidies are conducive to reducing investment costs, risks to enterprises, stimulating enterprises to invest in R&D, and giving full play to the positive externalities of R&D investment. Thereby enhancing the R&D and innovation capabilities of the entire industry. Klette and Moen [18] argue that R&D activities have typical externalities. That is, the knowledge spillovers generated by corporate R&D activities will enable other companies, including competitors, to gain knowledge sharing, R&D, and innovation capabilities. From the perspective of factor endowment theory, Wang Jun [19] believes that government subsidies reduce the financial risk of enterprise technology input, which is conducive to the transformation of production methods. That uses technology research and innovation as the main production factor input, and stimulates the multiplier effect of technology innovation. Therefore, government R&D subsidy promotes R&D investment and R&D enthusiasm of the enterprise. From the perspective of signal transmission theory, Meuleman, Maeseneire, and Kleer found that a government R&D subsidy's receipt implies a significant market potential, thus servings as a clear signal. It also shows that the corporate brand has a good reputation, which is more likely to attract bank loans and social fund investment, reducing corporate financing constraints, thereby promoting private investment [20,21].

In the process of empirical research, the study of the "crowding-in" school also went through three stages, from the research focus on whether there is a promotion effect of government R&D subsidies, to the further exploration of the degree of subsidy promotion, and the study of the effect of government subsidies under different factors. Matthias Almus and Dirk Czarnitzki analyzed public R&D policy schemes' effects on firms' innovation activities in Eastern Germany. They investigate the average causal impact of all public R&D schemes in Eastern Germany using a non-parametric matching approach. Compared to the case in which no public financial means are provided, it turns out that firms increase their innovation activities by about four percentage points [22,23].

### 2.1.2. Crowding-Out Effect

Montmartin analyzes from the perspective of the squeeze-out effect theory. Under the premise of a certain total investment in research and development projects, government R&D subsidies will squeeze private funds out of the original advances, resulting in government funds being lost due to private investment reduction [24]. According to the theory of supply and demand, David et al. believe that government R&D subsidies will stimulate market demand for R&D factors [12]. Yufen Chen et al. believe that with constant supply, increased demand will lead to increased equilibrium prices. That is, government R&D subsidies will increase R&D factors. Price (such as raising the researcher's salary), which increases the company's R&D costs, reduces marginal revenue, and directly results in reduced corporate R&D expenditures [25]. According to rent-seeking theory, government R&D subsidies provide rent-seeking opportunities for specific privileged organizations and monopolies, leading to failure of government intervention. Liu Hong and others believe that enterprises, especially state-owned enterprises, obtain government R&D subsidies through rent-seeking. After receiving the subsidies, they may not be used effectively in their R&D activities, making government R&D subsidies meaningless [26].

Empirical research aspects of extrusion effect: in the first stage, scholars used different data samples to confirm through empirical research that government R&D subsidies may indeed have a crowding-out effect. For example, Wallsten used US data to draw a conclusion about the crowding-out effect of government R&D subsidies [9]. Catozzella and Vivarelli used Italian enterprises' data samples and finally concluded that government subsidies and enterprise input-output were negatively correlated without considering enterprises' heterogeneity [27]. Peng Hongxing and Wang Guoshun (2018) used the data of China's A-share high-tech listed companies from 2009 to 2014 to conduct empirical research using Ordinary Least Squares(OLS) and Propensity Score Matching (PSM) models. The study found that innovation subsidies significantly reduced the total factor productivity of high-tech companies, caused companies to overinvest, and increase employee redundancy [28]. Scholars in the second stage introduced the control variables of the crowding-out effect. For example, Montmartin and Herrera introduced two control variables of the government subsidy method and the subsidy rate. They studied 25 Organization for Economic Cooperation and Development (OECD) members from 1990 to 2009 through dynamic spatial panel data [24]. The data from China verified that both direct and indirect subsidies from the government would have a crowding-out effect on the R&D investment of enterprises. Zheng Shilin and Liu Hewang introduced the control variable of the proportion of government R&D subsidies. They found that it is difficult for special government funds to increase corporate R&D investment and labor productivity. The higher the ratio of government R&D funding, the lower the innovation input and labor productivity [29].

### 2.1.3. Non-Significant or Mixed Effect

The above two conclusions are contradictory, mainly due to differences in sample data, research objects, and variable selection. In recent years, more and more research conclusions show that the government R&D subsidy's impact on corporate investment is not a unilateral promotion effect or a unilateral crowding-out effect. Both effects exist simultaneously and appear differently depending on the circumstances. Guellec analyzed the data of 17 OECD member countries and found that government subsidies have a promotion effect in the short term (within 1 year) and the long term (beyond 4 years), but will have an extrusion effect in the medium term (about 3 years) [30]. Clausen [31] used Norwegian data, found that the government's "research subsidy" has a promoting role, while the "development subsidy" has a crowding-out effect. Estimates are obtained with a cross-section sample of Spanish firms. Isabel Busom's study shows that public funding induces more private effort, but for some firms (30% of participants) full crowding-out effects cannot be ruled out and firm size remains related to effort. Whether or not a firm gets public funding [32]. Dingding Xiao and others believe that Chinese government subsidies will have different effects in different regions. In the eastern region, government subsidies

have a leverage effect on enterprises' R&D investment, but they have more extrusion effects in the central and western regions [33]. The study of Shuang Wang, Shukuan Zhao, Dong Shao and Hongyu Liu aims to discuss government subsidies' incentive effect on enterprise innovation investment based on different enterprise ownership. Employing sample data of listed Chinese manufacturing companies between 2011 and 2019, the findings suggest that the intensity of government subsidies exerts an incentive effect on corporate innovation investment; however, the incentive effect is different under the influence of political connections and investor attention. In particular, political connections inhibit the incentive effect, and investor attention promotes the incentive effect [34]. Guo Yingfeng and others believe that there is no "inverted U-shaped" curve or "positive U-shaped" curve relationship between Chinese government subsidies and the R&D investment of enterprises themselves. [35].

Thus the impact of government subsidies on corporate investment is not purely a promotion effect or a crowding-out effect, but will be mixed, depending on the timing of government subsidies, regions, and other factors. Table 1 summarizes the main representative views in the above-mentioned literature.

**Table 1.** Summary of significantly advanced research.

| Author (Time) | Research Object | Main Points |
|---|---|---|
| Czarnitzki D, Fier A (2002) | Public capital investment in Germany | Public capital does not have a crowding-out effect on company research and development (R&D) investment. |
| Callejón, García-Quevedo (2005) | Panel data in Spain | The results suggest that public subsidies have complemented private R&D |
| Wolff, Reinthaler (2008) | Organization for Economic Co-operation and Development(OECD) member data | Government subsidies can bridge the gap between private benefits and social benefits due to the externalities of R&D activities, and increase the enthusiasm of corporate R&D activities. |
| Wallsten (2000) | US data | Confirm the crowding-out effect of government subsidies. |
| Montmartin, Herrera (2015) | Data of the 25 OECD member countries, 1990–2009 | Both direct and indirect government subsidies will have a Crowding-out effect on corporate R&D investment. |
| Lee (2011) | Unique firm-level data for nine industries across six countries | These multiple channels indicate that it is difficult to evaluate the aggregate effect of public R&D support and that there are differential effects of public R&D support on firm R&D, depending on the various firm- or industry-specific characteristics. |
| Xiao Dingding (2013) | The provincial panel data of China from 1997 to 2008 | Chinese government subsidies will have different effects in different regions. In the eastern region, government subsidies have a leverage effect on enterprises' R&D investment; however, they have more Crowding-out effect in the central and western areas. |
| Li Ling, Tao Houyong (2013) | Data of 974 Chinese listed companies in 2010 | Government subsidies have a significant impact on the R&D investment of private enterprises, but have no significant effect on the R&D investment of state-owned enterprises. |
| Guo Yingfeng (2016) | Panel data of China's large and medium-sized industrial enterprises from 2004 to 2015 | There is no "inverted U-shaped" curve or "positive U-shaped" curve relationship between Chinese government subsidies and the R&D investment of enterprises |
| Shuang Wang, Shukuan Zhao, Dong Shao and Hongyu Liu (2020) | Data of listed Chinese manufacturing companies between 2011 and 2019 | The intensity of government subsidies exerts an incentive effect on corporate innovation investment; however, the incentive effect is different under the influence of political connections and investor attention. |

*2.2. Differentiation from Existing Research*

Increasing attention has been paid to the impact of government subsidies on technological innovation. However, the studies have not yet reached a unified conclusion. After reviewing the relevant literature, we find that scholars mainly study the impact of government subsidies on enterprise innovation from a single perspective, such as the size of the enterprise, the type of enterprise R&D investment, industry characteristics, and the time lag of policy effects, but the theoretical framework of how government subsidies promote enterprise technological innovation has not yet been formed. Most of the research objects in the existing research are manufacturing companies, and there is little attention to the emerging information and communication technology industry. In this study, we take the listed companies in China's information transmission, software and information technology service industries from 2012 to 2018 as the research object, and incorporate the macro-institutional environment, the meso-level market structure, and micro-corporate governance using the theoretical framework to establish the relationship between government subsidies and corporate technological innovation. This enhances a greater understanding of the theoretical frameworks behind government subsidies for technological innovation.

## 3. Research Hypotheses and Research Design

Government subsidies are an important part of fiscal expenditures. They are the transfer of free funds directly or indirectly provided by the government to microeconomic entities in accordance with the political and economic policies and policies of a certain period and according to specific purposes [36–38]. In 2014, the State Council of the People's Republic of China No. 11, 'Several Opinions of the State Council on Improving and Strengthening the Central Government's Scientific Research Projects and Fund Management' clearly stated: financial technology subsidies should focus on basic frontiers, public welfare, market-oriented and major projects [39], and local governments will issue them accordingly A wide variety of science and technology subsidy policies have been introduced. Government subsidies include financial appropriations, financial discounts, tax rebates, and non-monetary asset allocations free of charge. The most common form of subsidies is financial appropriations [40]. Government R&D subsidies mainly include subsidies to promote innovation activities, subsidies to promote enterprise development, subsidies to promote financing, subsidies to promote innovation and culture, and subsidies to promote talent accumulation [41]. Chinese-style two-tier subsidies are subsidies for enterprises by the central government and local governments at the same time. The central government subsidies are mainly for central enterprises or national-level encouragement and support enterprises or projects, and the local government subsidies are mainly for local enterprises and local enterprises or enterprises that encourage development and support [42]. This research is based on the impact of all the R&D subsidies in place by the state and local governments that the enterprise obtains on the enterprise's innovative R&D investment.

*3.1. Research Hypotheses*

The main regression analysis of this study is to study the impact of government subsidies on enterprise innovation. In addition, this article attempts to construct a theoretical framework for analyzing how government subsidies promote enterprise technological innovation, hoping to explore the theoretical mechanism of government subsidies on enterprise technological innovation from three levels of macro-institutional environment, meso-market structure and micro-corporate governance. It is hoped that the research conclusion will be able to explain how to better promote the leverage effect of government subsidies on enterprise innovation from three different levels in the context of China. Therefore, this paper proposes hypotheses from the following four aspects.

### 3.1.1. Government Subsidies and Investment in Technological Innovation

Studies provide different views on the role of government subsidies in enhancing innovation. Scott performed the ordinary least square using the enterprise R&D data as the dependent variable and government R&D input as the independent variable. They obtained a positive relationship, revealing that government subsidies encourage enterprises to increase technological innovation [43]. Levin and Reiss used panel data of the U.S. industry to combine structural variables such as industrial market concentration, asset specificity, R&D, advertising, and technical opportunities to form structural equations and use two-stage least squares regression. The results show that government subsidies can encourage enterprises to increase investment in innovation [44]. Data from 2005 through 2007 were obtained from a survey of small and medium-sized enterprises (SMEs) in biotechnology in South Korea. Kyung-Nam Kang, Hayoung Park find the government support through project funding directly and indirectly affects firms' innovation by stimulating internal R&D and domestic upstream and downstream collaborations [45].

In the Chinese context, many studies agree that government subsidies affect technological innovation. Hu investigate the high-tech companies in the Haidian district in Beijing in 1995 and found that government subsidies significantly promoted technological innovation [46]. Xie Weimin et al. sampled different companies from 2003 to 2005 to examine the relationship between government funding and the enterprises' technological innovation behavior. They concluded that government funding has a significant positive correlation to innovation on the listed companies [47]. Di Guo, Yan Guo, Kun Jiang examine the effects of the Innovation Fund for Small and Medium Technology-based Firms (Innofund). Using a panel dataset on Chinese manufacturing firms from 1998 to 2007, they find that Innofund backed firms generate significantly higher technological and commercialized innovation outputs than their non-Innofund backed counterparts and the same firms before winning the grant [48].

In an econometric study summarizing the impact of existing R&D subsidies on private R&D expenditures, the 'crowding-in hypothesis' accounts for approximately 60%, and the other two hypotheses account for approximately 20% each [17]. Based on these studies, this paper proposes Research Hypothesis 1:

**Hypothesis 1.** *Chinese government subsidies can promote enterprises' technological innovation input.*

### 3.1.2. Marketization Index of China's Provinces, Government Subsidies, and Investment in Enterprises Innovation

Chen Zongsheng proposed that marketization plays a pivotal role in the market mechanism and allocates free resources [49]. The economy's dependence on the market mechanism has increased, and the market mechanism has evolved from gradual emergence to maturity. To study the effect of government subsidies on technological innovation, the market characteristics where the enterprise is located must be considered. Previous studies agree that marketization can help enterprises to increase investment in innovation. Heitor and Murillo suggest that in regions where marketization is higher, the market can pass credit commitments to enterprises through competitions. This stimulates enterprise technology innovation and prevents companies from engaging in simple technology imitation [50]. Dittmar et al. [51] Pin Kowitz et al. [52], based on empirical studies of cross-border samples, found that in countries with low investor protection levels, corporate cash holdings are characterized by high levels and low value. Research by Frésard et al. [53] shows that, thanks to the effective regulatory mechanism in the United States, compared with the companies listed only in the mainland, the companies listed in the mainland and the United States have higher value of excess cash holdings.

The empirical literature for countries in transition pointed out that even when property rights are not privatized, market-oriented economic reforms may positively impact state-owned enterprises' operating efficiency. For example, Pinto et al. found that with the liberalization of price control, the intensification of corporate competition, and the

hardening of budget constraints, Polish state-owned enterprises' operating performance improved significantly [54]. Li [55] 272 state-owned enterprises in 1989 as an example, it was found that with the introduction of China's economic reforms, the total factor productivity (TFP) of state-owned enterprises has improved significantly. Falcetti et al. found that with the deepening of economic liberalization and property rights reform in Eastern European economies in transition, its impact on economic growth gradually appeared [56]. Fang Junxiong found that when marketization is high, capital transfers from low-efficiency areas to high-efficiency areas faster. That is, capital allocation is further optimized [57]. Fan Gang, Wang Xiaolu, and Ma Guangrong's research on China show that from 1997 to 2007, the marketization index to economic growth reached an average of 1.45 percentage points per year. The advancement of the market-oriented reform process has improved the efficiency of resource allocation and microeconomic efficiency. During this period, 39.23% of the total factor productivity growth was contributed by market-oriented reforms [58].

This study argues that the degree of government intervention in enterprises, the determination of product prices, and the protection of intellectual property are important indicators of marketization. In areas where the marketization index in China's provinces is high, enterprises will win more market shares and provide better products and services. If the government subsidies cannot fully meet technological innovation's needs, the enterprises will actively increase their investment in technological innovation to develop new products for market needs. This helps them to achieve competitive advantages. If the intellectual property is adequately protected, enterprises can obtain excess profits in innovation and technology transfer. In this case, the government subsidy can greatly promote technological innovation. Therefore, this paper proposes the following research Hypothesis 2:

**Hypothesis 2.** *The higher the level of marketization, the more significant the Chinese government subsidy in promoting the enterprise's technological innovation.*

### 3.1.3. Product Market Competition, Government Subsidies, and Enterprise Technology Innovation

The market competition includes competition intensity and competition strategy. In empirical research, product market competition refers to the intensity of market rivalry.

The relationship between product market competition and technological innovation has always been the focus of debate among scholars. Competition influences innovation in two ways. The first is the "Schumpeter effect". Many capital sources are mainly from internal enterprise financing, and competition will reduce firms' excess profits. However, monopoly strengthens the inherent motivation and allows firms to carry out innovative R&D [59]. Fierce product market competition means that companies may face risks such as insecure market position, loss of market share, and even bankruptcy [60]. In this case, companies may focus on survival, adopt conservative business strategies, and reduce the willingness to take risks and the motivation for active innovation [61]. Research by Xia Qinghua and others found that excessive product market competition hinders the realization of product value, increases business operating costs, and is not conducive to improving business performance. The corporate performance will affect the continuity of corporate innovation investment [62]. Lu Xiaomeng et al. found product market competition based on the industrial enterprise database using a fixed-effect model. As a result, the company's external financing cost has increased, and the unknown risks have increased [63]. The company will choose to reserve a large amount of cash instead of investing cash in innovative projects with no short-term return [64].

The second is to escape competition. With innovation, a company has an inherent capability to escape competition. Scherer pointed out that monopoly will lead to organizational inertia, and competition will bring stronger innovation motivation to enterprises [65]. The more intense the competition, the more the company strengthens its core competitiveness through R&D, the more obvious the effect of government subsidies. Conversely, when companies face no competition, they will not perform high-risk R&D to achieve a

monopoly. In such cases, government subsidies will allow companies to create price advantages by reducing costs, rather than investing in R&D. In this, investment in corporate R&D cannot be promoted with government subsidies in R&D. Jian Ze and Duan Yongrui's empirical research based on the data of Chinese industrial enterprises also reached a similar conclusion. Competition has a significant role in promoting technological innovation [66]. In the follow-up research, it is found that the "Schumpeter effect" and the "competition escape effect" can exist at the same time, and the relationship between product market competition and enterprise technological innovation is not a simple linear relationship [67]. Nie Huihua and other empirical studies based on the data of Chinese industrial enterprises found that competition is conducive to promoting enterprise innovation to a certain extent. In contrast, excessive competition will weaken enterprises' profit accumulation and reduce their technological innovation capabilities [68].

Research on government policy formulation to support enterprise technological innovation believes that it is precisely because of the significant externalities of innovation that only product market competition to allocate resources can easily lead to market failure. Enterprises do not have a strong desire for technological research and development, so governments of all countries spare no effort to locally formulate policies that encourage enterprises to carry out technological R&D and innovation [69], and promote the improvement of innovation capabilities [17]. In a developing country such as China, the marketization is in transition and is significantly different from developed countries. In the context of China, the effect of escaping competition is dominant, and we propose Hypothesis 3:

**Hypothesis 3.** *The more competitive the product market, the more successful the government subsidies in promoting enterprise technology innovation.*

### 3.1.4. Ownership Concentration, Government Subsidies, and Enterprise Technology Innovation Input

In the current competitive business environment, firms must improve their technological innovation capabilities. Due to high risks and long investment cycles, technological innovation creates an information imbalance between shareholders and managers. The agency problem is severe, and this directly affects the degree of enterprise participation in technological innovation. Wright et al. believe that the existence of agency problems causes managers to mainly care about personal wealth, job security, power prestige, and the maximization of personal utility, which will seriously affect and weaken their pursuit of innovation [70]. Jensen and Meckling (1976) believe that through the implementation of equity, stock options, and other incentive mechanism arrangements linked to operators' current performance, operators and owners' interests can be aligned, which can effectively improve operators' support for technological innovation [71].

An enterprise's operation requires a set of basic systems, namely corporate governance, whose purpose is to ensure the enterprise's sustainable development. Therefore, corporate governance is an institutional basis for enterprise technological innovation [72,73]. The shareholding structure determines the most basic governance structure of a company. Various problems in the company's development and corporate governance can be rooted at the shareholder level [74]. As a property right arrangement that determines the distribution of power and benefits within a company, the ownership structure is an essential factor that affects corporate R&D investment. Especially for China in an economic transition period, the property rights system is not yet complete, and the development of the market economy is not yet mature. As an important factor influencing innovation behavior, the property rights factor has special research significance [75]. The economics of property rights developed by Coase [76], Alchain [77], Demsetz [78], Cheung [79] and North [80] emphasized that property rights and the institutional environment play an important decisive role in economic behavior. Jefferson's research shows that the market environment and the nature of property rights have an important influence on enterprises' innovation behavior [81].

The existing research on the relationship between ownership structure and corporate innovation can be roughly summarized in four stages [82]. In the first stage, based on developed countries' research background, scholars believe that equity concentration can provide an effective supervision mechanism [73], which can solve the agency conflict between shareholders and managers caused by equity dispersion and the resulting insider control problems [83]. Therefore, increasing the equity ratio of business owners can promote the improvement of the level of R&D investment [84,85]. In the second stage, based on a comparative analysis based on the institutional differences and the rule of law environment between developed and developing countries [86], scholars found that due to the weak system and rule of law environment under the new economic system, the concentration of corporate equity may lead to the dual agency problem [87]. The willingness and ability to control shareholders to pursue private interests is enhanced [88]. The company's resources can be transferred through complex related transactions, which leads to insufficient allocation of innovation resources [89]. Chin et al. [90] through empirical research found that when equity is over-concentrated, due to the unity of investment by major shareholders and the high risk of innovation, major shareholders will exhibit a certain risk aversion to prevent damage expected short-term benefits. Therefore, the company's investment in innovation is inhibited. In the third stage, when the existing empirical research shows the positive or negative relationship between the ownership structure and technological innovation, some scholars believe that the degree of ownership concentration does not significantly improve innovation performance [91]. Due to the research results' inconsistency, in the fourth phase of the study, scholars began to question the linear relationship between equity concentration and corporate innovation. They believed that the effect of equity concentration on innovation performance was not a simple linear relationship, but an inverted U-shape relationship [92–94].

This paper believes that improvement in innovation must rely on continuous investment in enterprise technological innovations. Since technological innovation is risky and characterized by long investment cycles, shareholders who invest in this enterprise will be more cautious when making investment decisions. Shareholders whose investments are higher in this company will carry more risk than those with lower investment. This will influence managerial decisions and affect the company's investment decisions in technological innovation. A company with a good governance mechanism can solve the agency problem, thereby increasing technological innovation investment. Based on the special shareholding structure of Chinese enterprises, on the one hand, the state-owned nature of a large number of legal person shares in China makes it more representative of the characteristics of the role of state shareholders [95]. In addition, the equity of Chinese private listed companies is also relatively concentrated. The phenomenon of "key person control" in private listed companies with family holding as the main feature is common, and the degree of equity balance is relatively low.In fact, a "dominant share" equity structure has been formed. This kind of shareholders' connected nature is not conducive to improving corporate governance efficiency and standardizing the company's operating behavior. The concentration of equity will affect the company's R&D investment behavior has a significant impact [75]. This paper proposes research hypothesis 4:

**Hypothesis 4.** *The more dispersed the shareholding, the more the Chinese government subsidy will promote the company's technological innovation investment.*

The analytical framework of the research is shown in Figure 1 below.

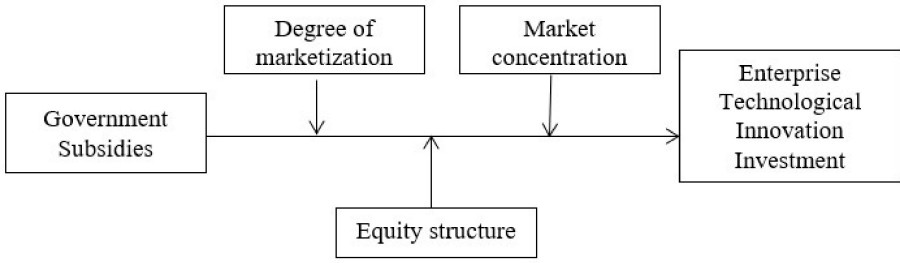

**Figure 1.** Analysis framework.

*3.2. Research Design*

Based on the above analysis, the model constructed in this paper is as follows. Model (1) examines the relationship between government subsidies and enterprise technological innovation inputs, model (2) examines the influence of marketization on the leverage of government subsidies, and model (3) examines the degree of market competition on the government. For the impact of the leverage of subsidies, model (4), examines the impact of equity concentration on government subsidies' leverage.

3.2.1. Model Construction and Variable Selection

The model is constructed as follows.

$$R\&D = \beta_0 + \beta_1 sub + \alpha_1 Size + \alpha_2 Debt + \alpha_3 Growth + \alpha_4 ROE + \alpha_5 SOE + \varepsilon \quad (1)$$

$$R\&D = \beta_0 + \beta_1 sub + \beta_2 Market + \beta_3 sub \times Market + \alpha_1 Size + \alpha_2 Debt + \alpha_3 Growth + \alpha_4 ROE + \alpha_5 SOE + \varepsilon \quad (2)$$

$$R\&D = \beta_0 + \beta_1 sub + \beta_2 HHI + \beta_3 sub \times HHI + \alpha_1 Size + \alpha_2 Debt + \alpha_3 Growth + \alpha_4 ROE + \alpha_5 SOE + \varepsilon \quad (3)$$

$$R\&D = \beta_0 + \beta_1 sub + \beta_2 PFIVES + \beta_3 sub \times PFIVES + \alpha_1 Size + \alpha_2 Debt + \alpha_3 Growth + \alpha_4 ROE + \alpha_5 SOE + \varepsilon \quad (4)$$

where R&D is the explained variable, which indicates the company's technological innovation; sub indicates the number of government subsidies received by the listed company in the current period; Market indicates the Marketization index of China's provinces; Herfindahl-Hirschman Index(HHI) indicates the product market competition; Shareholding ratio of the top five shareholders(PFIVES) indicates the concentration of equity, and size, debt, growth, return on equity (ROE) and state-owned enterprise (SOE) indicate a series of control variables. The specific definitions of the variables are shown in Table 2. This paper studies the impact of government subsidies on enterprise R&D investment, and draws on the existing literature (Matthias Almus [22], Kyung-Nam Kang [45], Liu Hong [26], Zhang Jie [38], Lu Xiaojun [3]). The year (Y) and industry (Ind) control in the research.

The dependent variable is the intensity of R&D investment. Regarding the selection of agency variables for enterprise technology innovation, this study uses the R&D investment to characterize its technology innovation. This is measured by the ratio of R&D investment to current operating income, disclosed in the listed companies' annual report.

The independent variable Sub represents government subsidies. As no particular government subsidy data exists, this study uses the "Subsidize revenue" in the listed companies' profit statements as an alternative, which includes explicitly fiscal subsidies, fiscal consolidation, new product returns, innovation incentives, and other government subsidies.

The moderate variable represents variables like level of marketization, product market competition, and Ownership Concentration.

MARKET represents the marketization index of China's provinces. This quantitatively determines that the degree of economic marketization is a too complicated task. Wang Xiaolu and Fan Gang studied the internal mechanism and influencing factors of the marketization index of China's provinces based on relevant marketization measurement systems at home and abroad [96]. China's marketization index consists of five aspects, each reflecting a part of marketization. They are the relationship between the government and the market, the development of the non-state-owned economy, the degree of development

of the product market, the degree of development of the factor market, the development of market intermediary organizations, and the rule of law. The market-oriented index consists of 18 fundamental indexes. The five market-oriented indexes are synthesized from the sub-indexes according to the equal weight calculation (the arithmetic mean). The total marketization index is composed of five according to the equal weight principle. Many scholars such as Liu Jianghui, Tang Dongbo [97], Li Zengquan, Liu Fengwei, Yu Xuhui [98], Zhou Fangwei, Yang Jidong [99] have used this indicator to study China's economic problems. This study uses the comprehensive report data from 2012 to 2018, a variable that constitutes China's provinces' marketization index.

**Table 2.** List of main variable definitions.

| Types | Names | Symbols | Definition |
|---|---|---|---|
| Dependent Variable | R&D investment intensity | R&D | Current enterprise R&D investment/operating revenue |
| Independent Variable | Government subsidy | Sub | Total government subsidies/total assets at the end of the period |
| Moderate Variable | Marketization index of China's provinces | Market | Greater than the median is defined as 1 and less than the median is defined as 0 |
| | Product market competition | HHI | Herfindahl-Hirschman Index (HHI index) |
| | Ownership Concentration | PFIVES | The sum of the shareholding ratios of the top 5 major shareholders of the company |
| Control Variable | Firm size | Size | The natural logarithm of the company's period-end assets |
| | Leverage | Debt | Total Liabilities/Total Assets |
| | Corporate growth | Growth | Growth rate of total assets |
| | Profitability | ROE | Return on Equity, Net income/owner's equity |
| | State-ownership | SOE | SOE = 1 if a state-owned firm, and 0 otherwise |

HHI represents Product market competition. This article uses the Herfindahl–Hirschman Index (HHI Index) to measure product markets' competitiveness based on existing research and practice.

PFIVES represents Ownership Concentration. At present, many scholars use the shareholding ratio of the top five shareholders to indicate the degree of Ownership Concentration (Zhang Hongjun [100]; Feng Genfu, [101]; Sun Zhaobin [102]). This article chooses these concepts to represent the degree of Ownership Concentration.

Control represents various control variables, other factors that may have an impact on the company's technological innovation, including the size of the company (Size), the asset-liability ratio (Debt), the growth of the company (Growth), profitability (Return on Equity, ROE), and equity attributes (state-owned enterprise, SOE).

SIZE indicates enterprise size, generally believed that large enterprises and small enterprises have different characteristics and possess distinct advantages in enhancing technological innovation. Large enterprises mainly have resource advantages, whereas small enterprises mostly have flexibility advantages in technological innovation. Debt indicates the asset–liability ratio. Low leverage capital structure helps companies to increase their investment in technological innovation, and high-debt companies will be more cautious about increasing their investment in technological innovation. The growth represents corporate growth. Alex Coada et al. used the panel vector autoregressive model to study the relationship between an enterprise's technological innovation and growth [103]. They found that the willingness of the growth companies to increase technology innovation is higher than the low-growth companies. ROE indicates profitability. Wang Renfei suggests that the higher the enterprise's profit rate, the higher the proportion of its investment in technological innovation [104]. Typically, a large scale of investment in technological innovation activities requires the support of excess profits. SOE represents property rights. Listed companies are classified into state-owned enterprises and non-state-

owned enterprises according to the nature of the largest shareholder's property rights. Gao Hongwei [105], Zhang Xinglong [106], Li Ling and Tao Houyong [10] have proved that government subsidies promote R&D investment in non-state-owned enterprises. However, due to insufficient incentives for the innovation of state-owned property rights, government subsidies have no induction effect on state-owned enterprises' R&D investment.

### 3.2.2. Selection of Samples and Acquisition of Data

In the twelfth "Five-Year Plan" from 2011, the Chinese government proposed "strengthening enterprises' dominant position in technological innovation and guiding innovation resources such as funds, talent, and technology to gather in enterprises." After the plan's formulation, the financial sector specifically increased subsidies on R&D. Zhang Jie proposed factors responsible for corporate monopolies in the current Chinese scenario because state-owned enterprises enjoy monopoly power, protected by government policies [107]. This is ineffective for innovative R&D activities. In order to avoid this problem, this article screened the research objectives. Lu Xiaojun analyzed China's strategic emerging industries' current status and pointed out that the proportion of state-owned capital and private capital varies among industries [4]. Among them, China's new-generation information technology industry is less affected by government intervention, and private capital has become its main force for development. According to the 2012 industry classification standard of China Securities Regulatory Commission, this paper identifies the listed companies of the information transmission, software, and information technology service industry as the research object according to the 2012–2018 panel data after the promulgation of the national industrial policy. We deleted companies listed after 2012, exclude ST companies and companies with incomplete data. We performed a 1% extreme value shrinking on continuous variables at the company level, a total of 146 companies that met the requirements were selected, and 1022 observations were obtained. We collected data from the China Economic and Financial Research Database and a corporate annual report published by Juchao Information Network. We used STATA version 16.0 for the statistical analysis. The industry, property rights, and regional distribution of the sample listed companies are shown in Tables 3 and 4, and Figure 2.

By describing the basic situation of the sample companies, the proportion of non-state-owned enterprises in the sample companies is high (82.9%), indicating that private capital in this industry is relatively active; whereas in the regional distribution, among the 31 provinces in China, only Beijing, Guangdong, Zhejiang, Shanghai, and Jiangsu account for 74%, indicating that the industry is mainly concentrated in the capital and the highly market-oriented eastern coastal areas.

**Table 3.** Industry distribution of sample listed companies.

| Industry Name | Number of Enterprises | Proportion |
|---|---|---|
| Software and Information Technology Services | 107 | 73.3% |
| Internet and related services | 32 | 21.9 |
| Telecommunications, radio, television and satellite transmission services | 7 | 4.8% |

**Table 4.** List of the property right of sample listed companies.

| Property Rights | Number of Companies | Proportion |
|---|---|---|
| State-owned enterprise | 25 | 17.1% |
| Non-state-owned enterprises | 121 | 82.9% |

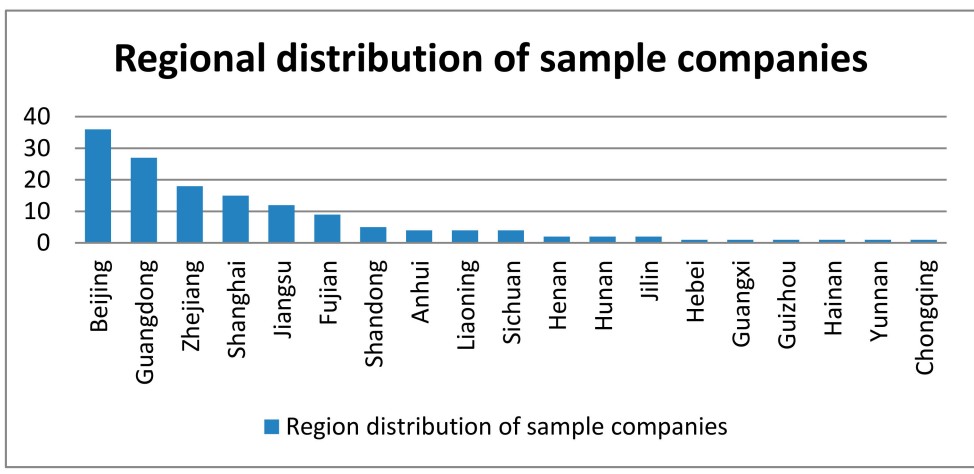

**Figure 2.** Regional distribution of sample companies.

## 4. Analysis Results

### *4.1. Statistical Analysis*

#### 4.1.1. Descriptive Statistics

Table 5 shows the descriptive statistical results of the main variables. Regarding the R&D intensity, the average value is only 0.96%, indicating that the sample company's R&D investment is still low. There is a large gap between enterprises' R&D investment intensity, with the minimum (0.0877%) and maximum (4.56%).

**Table 5.** Descriptive statistics of the main variables.

| Variable | Obs | Mean | Std. Dev. | Min | Max |
|---|---|---|---|---|---|
| Sub | 1022 | 0.0096121 | 0.0091691 | 0.0000877 | 0.0456307 |
| Market | 1022 | 8.857278 | 1.218248 | 4.81 | 10.34242 |
| HHI | 1022 | 0.1257136 | 0.191635 | 0.032737 | 0.890434 |
| PFIVES | 1022 | 0.4827531 | 0.1351321 | 0.209436 | 0.756872 |
| Size | 1022 | 21.56564 | 1.132379 | 7.607381 | 24.34312 |
| Debt | 1022 | 0.3089441 | 0.1712771 | 0.038461 | 1.412498 |
| Growth | 1022 | 0.2767396 | 0.4829885 | −0.527871 | 2.586532 |
| ROE | 1022 | 0.0688537 | 0.1346278 | −1.868176 | 1.610631 |
| SOE | 1022 | 0.1741683 | 0.3794398 | 0 | 1 |

#### 4.1.2. Correlation Coefficient Test

From Table 6, the correlation coefficient between Chinese government subsidies and enterprises' technological innovation is positive.

**Table 6.** Correlation coefficient test of main variables.

| | R&D | SUB | Size | Debt | Growth | ROE | SOE |
|---|---|---|---|---|---|---|---|
| R&D | 1 | | | | | | |
| Sub | 0.285 | 1 | | | | | |
| Size | −0.0771 | −0.0995 | 1 | | | | |
| Debt | −0.235 | −0.108 | 0.295 | 1 | | | |
| Growth | −0.0916 | −0.0546 | 0.0570 | −0.0643 | 1 | | |
| ROE | −0.0110 | 0.148 | 0.0612 | −0.156 | 0.297 | 1 | |
| SOE | −0.186 | 0.00530 | 0.210 | 0.205 | −0.131 | 0.0163 | 1 |

#### 4.1.3. Univariate Test

As shown in Table 7, through the mean test, the impact of Chinese government subsidies on different group companies' technological innovation is significantly different.

**Table 7.** Group test by the average value of government subsidies.

|  | Obs | Mean | Mean *t* Test |
|---|---|---|---|
| Sub < Mean | 650 | 0.088 | −7.6651 *** |
| Sub > Mean | 372 | 0.131 | |

Note: The *t*-test is used for the mean, and *** indicating that the test passed at the 1% significance level.

*4.2. Empirical Regression Analysis Results*

4.2.1. Main Effect Regression Analysis

In Table 8, column (1) does not add any control variable, column (2) adds the main control variable, and column (3) adds the main control variables, year, and industry.

**Table 8.** Regression analysis table of government subsidy and enterprise technology innovation input.

| Variable | (1) | (2) | (3) |
|---|---|---|---|
| Constant | 0.078 *** | 0.034 | 0.044 |
| | (9.82) | (0.46) | (0.51) |
| Sub | 2.760 *** | 2.635 *** | 2.652 *** |
| | (4.13) | (4.02) | (4.07) |
| Size | | 0.004 | 0.004 |
| | | (1.23) | (0.90) |
| Debt | | −0.104 *** | −0.103 *** |
| | | (−3.59) | (−3.57) |
| Growth | | −0.018 *** | −0.019 *** |
| | | (−3.57) | (−3.42) |
| ROE | | −0.036 * | −0.034 |
| | | (−1.67) | (−1.55) |
| SOE | | −0.040 *** | −0.039 *** |
| | | (−3.40) | (−3.30) |
| Year/Ind | NO | NO | YES |
| N | 1022 | 1022 | 1022 |
| Adjusted R2 | 0.080 | 0.157 | 0.153 |
| F | 17.04 | 12.45 | 8.16 |

Note: The data are the regression coefficients of the respective variables, and the bracketed values are the revised T values; ***, * indicate statistical significance at 1% and 10%, respectively.

A significant positive correlation is recorded from the above regression results between government subsidy SUB and enterprise technology input ($\beta = 2.652$, $\rho < 0.01$), that is, government subsidy can promote enterprise technology innovation input. Thus, Hypothesis 1 is verified.

Among the controlled variables, the company size (SIZE) has no significant effect on its technological innovation, indicating that the correlation between the company size and the R&D intensity is not significant. The asset-liability ratio (DEBT), growth capability (GROWTH), ROE, and property rights (SOE) of the enterprise are negatively correlated to the strength of R&D investment, indicating that the stronger the company's debt-paying ability, the higher the strength of R&D investment. Simultaneously, the more companies lag, the stronger the willingness to innovate in R&D, the greater the investment. The enterprise will actively increase investment in technological innovation to reduce the competition and to achieve competitive advantages. To improve its profitability, the "escape from competition effect" is noticeable. Moreover, non-state-owned enterprises have a higher investment in technological innovation than state-owned enterprises.

4.2.2. Analysis of the Moderating Effects of Marketization Process, Market Competition, and Ownership Concentration

To verify Hypothesises 2–4, the intensity of R&D investment was used as a proxy variable for R&D. We examined whether the effects of Chinese government subsidies are different on the enterprise's technological innovation in industries that compete in different markets and enterprises that compete with varying concentrations of ownership.

The regression results are shown in Table 9. Columns (1) and (2) are the adjustment of the marketization process, of which column (1) does not control year effect and industry effect, column (2) controls the year effect and industry effect; columns (3) and (4) are the adjustment of the degree of market competition. Where column (3) does not control the year effect and industry effect, column (4) controls the year effect and industry effect; columns (5) and (6) are the adjustment of the degree of Ownership Concentration, where column (5) does not control the year effect and industry effect, column (6) controls the year effect and industry effect.

**Table 9.** Analysis of the regulatory effects of marketization process, market competition and Ownership Concentration.

| 9 | (1) | (2) | (3) | (4) | (5) | (6) |
|---|---|---|---|---|---|---|
| Cons | 0.059 | 0.058 | 0.023 | 0.018 | 0.051 | 0.045 |
| | (0.80) | (0.69) | (0.31) | (0.20) | (0.64) | (0.50) |
| Sub | 1.387 ** | 1.397 ** | 2.796 *** | 2.798 *** | 7.697 *** | 7.697 *** |
| | (2.02) | (2.05) | (3.45) | (3.46) | (3.11) | (3.10) |
| Market | −0.027 ** | −0.026 ** | | | | |
| | (−2.35) | (−2.20) | | | | |
| Sub×Market | 3.316 *** | 3.324 *** | | | | |
| | (2.95) | (2.93) | | | | |
| HHI | | | −0.038 | −0.039 | | |
| | | | (−1.65) | (−1.60) | | |
| Sub×HHI | | | −2.749 | −2.726 | | |
| | | | (−0.83) | (−0.82) | | |
| PFIVES | | | | | 0.028 | 0.026 |
| | | | | | (0.47) | (0.44) |
| Sub×PFIVES | | | | | −10.347 ** | −10.351 ** |
| | | | | | (−2.27) | (−2.25) |
| Size | 0.004 | 0.004 | 0.005 | 0.005 | 0.003 | 0.003 |
| | (1.05) | (0.89) | (1.42) | (1.21) | (0.91) | (0.86) |
| Debt | −0.108 *** | −0.107 *** | −0.106 *** | −0.106 *** | −0.106 *** | −0.106 *** |
| | (−3.79) | (−3.80) | (−3.69) | (−3.70) | (−3.56) | (−3.58) |
| Growth | −0.016 *** | −0.017 *** | −0.017 *** | −0.017 *** | −0.015 *** | −0.016 *** |
| | (−3.30) | (−3.21) | (−3.21) | (−3.10) | (−2.86) | (−2.80) |
| ROE | −0.045 ** | −0.046 ** | −0.0340 | −0.0350 | −0.0300 | −0.0310 |
| | (−2.12) | (−2.12) | (−1.56) | (−1.55) | (−1.37) | (−1.35) |
| SOE | −0.036 *** | −0.036 *** | −0.036 *** | −0.036 *** | −0.038 *** | −0.038 *** |
| | (−3.30) | (−3.24) | (−3.08) | (−3.04) | (−3.12) | (−3.09) |
| Year/Ind | NO | YES | NO | YES | NO | YES |
| N | 1022 | 1022 | 1022 | 1022 | 1022 | 1022 |
| Adjusted R2 | 0.183 | 0.179 | 0.172 | 0.167 | 0.185 | 0.180 |
| F | 11.20 | 8.075 | 13.38 | 8.894 | 11.31 | 8.116 |

Note: The data are the regression coefficients of the respective variables, and the bracketed values are the revised T values; ***, ** indicate statistical significance at 1%, 5%, respectively.

According to the regression results of columns (1) and (2) in Table 8, we find that the interaction between the marketization of China's provinces and government subsidies has a significant positive correlation with the intensity of enterprise R&D investment ($\beta$ = 3.324, $\rho$ < 0.01). This shows that marketization has a stimulating effect on government subsidies on corporate innovation. Hypothesis 2 passes the test.

Through the regression results of columns (3) and (4), we find that product market competition and government subsidies are negatively related to the intensity of R&D investment, but not significantly. The results show that the strength of product market competition in the sample industry has no significant effect on the promotion effect of government subsidies on enterprise R&D. Hypothesis 3 fails the test.

Through the regression results of columns (5) and (6), we find that the interactions of Ownership Concentration and government subsidies have a significant negative correlation

for the intensity of enterprise R&D investment ($\beta = -10.351$, $\rho < 0.05$). This shows that the more dispersed the corporate equity, the more government subsidies increase the intensity of corporate R&D. Hypothesis 4 passes the test.

Moreover, the test of the control variables from SIZE-SOE is consistent with the main effect of regression analysis.

### 4.2.3. Robustness Test, Replacing Variables

To test the robustness of the result, we performed the following test on the regression results, using the ratio of corporate R&D investment to total assets as the explained variable and the ratio of government subsidies to total operating income as the explanatory variable. The results are shown in Table 10. Column (1) is the main effect; column (2) is the adjustment of the marketization process; column (3) is the adjustment of the degree of market competition, and column (4) is the adjustment of the degree of Ownership Concentration. (1)–(4) Columns control the year effect and industry effect.

**Table 10.** Regression results of government subsidies and enterprises' technological innovation input (robustness test).

| Variable | (1) | (2) | (3) | (4) |
|---|---|---|---|---|
| constant | 0.0350 | 0.0360 | −0.00300 | 0.0490 |
|  | (0.40) | (0.42) | (−0.04) | (0.55) |
| Sub | 1.084 *** | 0.701 ** | 1.210 *** | 2.530 *** |
|  | (4.04) | (2.19) | (5.26) | (3.07) |
| Market |  | −0.0190 |  |  |
|  |  | (−1.61) |  |  |
| Sub×Market |  | 1.000 ** |  |  |
|  |  | (2.19) |  |  |
| HHI |  |  | −0.048 ** |  |
|  |  |  | (−2.26) |  |
| Sub×HHI |  |  | −0.900 |  |
|  |  |  | (−0.85) |  |
| PFIVES |  |  |  | −0.0150 |
|  |  |  |  | (−0.32) |
| Sub×PFIVES |  |  |  | −2.781 * |
|  |  |  |  | (−1.84) |
| Size | 0.00400 | 0.00400 | 0.00600 | 0.00400 |
|  | (0.89) | (0.99) | (1.33) | (0.91) |
| Debt | −0.082 *** | −0.086 *** | −0.084 *** | −0.086 *** |
|  | (−2.81) | (−2.94) | (−2.89) | (−2.93) |
| Growth | −0.020 *** | −0.020 *** | −0.018 *** | −0.017 *** |
|  | (−3.91) | (−3.84) | (−3.35) | (−3.24) |
| ROE | −0.0200 | −0.0270 | −0.0220 | −0.0170 |
|  | (−0.86) | (−1.18) | (−0.97) | (−0.72) |
| SOE | −0.032 *** | −0.031 *** | −0.028 ** | −0.031 ** |
|  | (−2.73) | (−2.67) | (−2.41) | (−2.55) |
| Year/Ind | YES | YES | YES | YES |
| N | 1022 | 1022 | 1022 | 1022 |
| Adjusted R2 | 0.181 | 0.202 | 0.204 | 0.206 |
| F | 6.899 | 7.098 | 9.574 | 7.054 |

Note: The data are the regression coefficients of the respective variables, and the bracketed values are the revised T values; ***, **, and * indicate statistical significance at 1%, 5%, and 10%, respectively.

The robustness test shows that the Chinese government subsidies and R&D investment intensity are significantly positive, supporting Hypothesis 1, and the other results are consistent with the previous analysis.

### 4.3. Analysis Conclusions

Chinese government subsidies can induce enterprises to increase technological innovation investment. This conclusion is the same as the current mainstream research

results [18–23]. The higher the marketization process index, the more significant government subsidies' incentive effect enterprise R&D investment. This conclusion is consistent because there is a large regional development gap in China's marketization process. It is also the same as the results of most scholars [50–58]. The industry concentration of the product market is negatively correlated to Chinese government subsidies on corporate technological innovation. Thus, this is not significant. The more dispersed the shareholding, the better the government subsidies promote the company's technological innovation. This conclusion is the same as that of Chin [90] and Luo Zhengying [75]. In the context of China's unique equity structure and environment, the more dispersed the equity, the more government subsidies will be used to promote technological innovation.

The failure of the hypothesis 3 test can be attributed to the following reasons. Zhang Jie [107] shows a significant positive relationship between competition and innovation under the Chinese scenario, showing that China's industry has the NN (Neck–Neck) structure type, and it is not developed like those of the United Kingdom and the United States. In the LL (Leader-Leader) structure type, the technology gap in China's industry is small. When there is a structure-type industry with a little technology gap between enterprises, competition has promoted corporate innovation. In the regression analysis, the coefficient of the interaction between HHI and SUB is negative, showing that the lower the market concentration, the better the government subsidies to promote the enterprise's technological innovation. From the results, the technology gap between companies in the industry is relatively small, and they belong to the NN (Neck-Neck) structure type. Therefore, the more intense the market competition, the more government subsidies will promote its technological innovation. However, the effect is not significant, owing to the lack of research samples.

## 5. Conclusions

As China is a country transitioning from a planned economy to a market economy, the Chinese government's industrial policy's impact on corporate decision-making behavior is high. A large gap in the degree of marketization is observed in different regions of China, and the degree of competition in various industries varies. Corporate governance problems caused by the excessive concentration of enterprise shares are still widespread. To this end, this paper uses 2012–2018 panel data of the listed companies in the information transmission, software, and information technology services industry to study whether R&D subsidies can make enterprises increase technological innovation and effectively leverage subsidies. The results found a significant positive correlation between Chinese government subsidies and enterprise technology input; that is, Chinese government subsidies have a significant role in promoting technological innovation in enterprises. The correlation of R&D investment intensity is significant in regions with higher marketization. Chinese government subsidies can increase the intensity of R&D investment. The industry concentration of the product market is negatively related to Chinese government subsidies on technological innovation. However, this is not significant. The interaction between Ownership Concentration and government subsidies has a significant negative correlation with the intensity of enterprise R&D investment.

In optimizing and upgrading the industrial structure, we must ensure that the market plays a decisive role and attach importance to the government's role, especially for positive externalities such as innovation. Based on the empirical conclusions, this study offers has the following suggestions for government subsidy policies.

Full leverage of government subsidies should be encouraged, and enterprises should increase investment in technological innovation. The Chinese government should promote market-oriented reforms that focus on fair market competition, create an excellent institutional environment, guide the optimal allocation of social resources, and offer endogenous forces to the market. To improve the independence of Chinese enterprises' innovation capabilities, and market competition should screen capable enterprises for government subsidies to promote technological innovation. The government can optimize enterprises'

equity structure by reducing barriers to entry of private capital and advancing the reform of mixed ownership of state-owned enterprises so that government subsidies can better promote investment in technological innovation.

This research takes the communication and information technology industry with high technological investment as the research object. The macro-institutional environment's three-dimensional perspective, meso-market structure, and micro-corporate governance establish a theoretical framework for the relationship between government subsidies and corporate technological innovation. The above makes people more aware of the theoretical basis for government subsidies for technological innovation.

Despite these contributions, this study has several limitations. First, this study only examined the impact of government subsidies on the enterprise's technological innovation. In a future study, we will examine government subsidies on enterprises' technological innovation's output and efficiency. In terms of intermediate variables, we only use the marketization process, product market competition, and Ownership Concentration to measure the macro-institutional environment, meso-market structure, and micro-company governance. Since there are many indicators to measure these three levels, this study chooses the three indicators that can best reflect its condition and ignores other indicators. In future research, we will enrich the measurement indicators to examine the results of the study. Only 146 Chinese listed companies were surveyed, leading to potential selection bias. Future research could expand the sample to address this problem.

**Author Contributions:** Conceptualization, L.J. and D.C.; methodology, L.J. and D.C.; software, L.J.; validation, L.J., E.N. and D.C.; formal analysis, L.J.; investigation, E.N.; resources, D.C.; data curation, L.J.; writing—original draft preparation, L.J.; writing—review and editing, E.N. and D.C.; visualization, L.J.; supervision, D.C.; project administration, E.N. and D.C.; funding acquisition, D.C. All authors have read and agreed to the published version of the manuscript.

**Funding:** This research was sponsored by Graduate School of Convergence Technology Innovation (Ministry of Trade, Industry and Energy, grant number P0012782).

**Institutional Review Board Statement:** Not applicable.

**Informed Consent Statement:** Not applicable.

**Data Availability Statement:** Publicly available datasets were analyzed in this study. This data can be found here: [https://cn.gtadata.com/].

**Conflicts of Interest:** The authors declare no conflict of interest.

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
