# Peer review of "Impact of Chinese Government Subsidies on Enterprise Innovation: Based on a Three-Dimensional Perspective"

_sustainability, doi:10.3390/su13031288_

Round 1

Reviewer 1 Report

Dear Authors,

I read the said paper with pleasure. It raises an interesting (and very actual) topic associated with the impact of subsidies on firm’s innovation. That’s why I evaluate the choice of the topic highly. The structure of the paper is also proper (with some minor comments - see below) as well as its language.

Despite these positive comments I am only partly satisfied with the manuscript. There are some aspects that need improvements (in some cases even substantial ones). First of all, literature review and hypotheses development. Unfortunately, for me this is the weakest part of the text and my final mark is mostly based on this aspect. The overview is rather chaotic. Being a mixture of the views of opinions of the scholars, presented in the unordered and messy way.

Secondly, hypotheses development is rather underdeveloped, being based on a very modest literature overview. I would rather merge Literature review and hypotheses development in one section; the current version causes that some hypotheses are formulated on the basis of 2-4 references (in one case this is just one reference). As a result to some extent there seem to me too obvious.

The literature also requires improvement. Its size is not so impressive as to journal’s of that level. Some papers on the topic – also from Sustainability journal – are missing, just to mention the following papers:

  • Shuang Wang, Shukuan Zhao, Dong Shao and Hongyu Liu (2020), Impact of Government Subsidies on Manufacturing Innovation in China: The Moderating Role of Political Connections and Investor Attention, Sustainability, 2, 7740; doi:10.3390/su12187740
  • Di Guo, , Yan Guo, Kun Jiang, (2016) Government-subsidized R&D and firm innovation: Evidence from China, Research Policy, 45(6), 1129-1144.
  • Kyung-Nam Kang, Hayoung Park (2012), Firm collaborations on innovation in Korean biotechnology SMEs, Technovation, 32(1), 68-78.

In addition, there are many local sources with rather limited number of studies of other scholars and in theoretical background one should present the wide overview of research results of the scholars dealing with the topic.

There is also lack of a Discussion section with reference to the results of other scholars (in line or in contrast). For sure it would enrich the paper which – I must admit it – presents really interesting results.

Furthermore, though the paper presents some implications (especially for business and policy makers), its contribution to the theory has not been presented at all.

Please also ensure in the whole paper the “from general to details approach’. Unfortunately, there are some places in the paper (for example in Introduction) in which this is omitted.

Best regards

Reviewer

Author Response

Dear Editor and Reviewers,

Thank you very much for your consideration of our manuscript and request for a revised version.

Please find the attached manuscript entitled as “Impact of Chinese government subsidies on enterprise innovation : Based on a three-dimensional perspective" to be reconsidered for publication at Sustainability.

We have copied and pasted all reviewers’ comments below, and addressed each one individually. As requested, we have listed the changes we have made including how and where. Please refer to the text marked as red for the listings of changes in the revised manuscript. As you will see, we have made every attempt to incorporate these suggestions as thoroughly as possible.

Yours sincerely,

Reviewer 2 Report

The presentation reflects the present state of knowledge. The text is easy to understand by scientists in other disciplines. The paper is very well structured. The Introduction section is good, in this section the authors presents clearly the objectives and the main contributions of the study. The authors had provided sufficient background and include relevant references. The research design is innovative and appropriate. The method is adequately described. The results are clearly presented. The conclusions are supported by the results.

Author Response

(The authors gave the same response as above.)

Reviewer 3 Report

The title is overly long. Why do authors split it in two parts using a semicolon?

There is a huge amount of literature relative to the relationship between subsidies/incentives and innovation. This is a list of some paper that authors can consider for improving their review analysis which results limited. See section 2 of the manuscript.

Almus, M.; Czarnitzki, D. The Effects of Public R&D Subsidies on Firms’ Innovation Activities. J. Bus. Econ. Stat. 2003, 21, 226–236.

Bellucci, A., Pennacchio, L. & Zazzaro, A. Public R&D subsidies: collaborative versus individual place-based programs for SMEs. Small Bus Econ 52213–240 (2019).

Busom, I. An empirical evaluation of the effects of R&D subsidies. Econ. Innov. New Technol. 2000, 9, 111–148.

Callejón, M.; García-Quevedo, J. Public Subsidies to Business R&D: Do They Stimulate Private Expenditures? Environ. Plan. C: Gov. Policy 2005, 23, 279–293.

Cao, Y.; Chen, M.; Su, G. Are fiscal and tax policies conducive to enhancing enterprise innovation efficiency? Southeast Acad. J. 2018, 2, 96–104.

Cappelen, A.; Raknerud, A.; Rybalka, M. The effects of R&D tax credits on patenting and innovations. Res. Policy 2012, 41, 334–345.

Chang, A.C. Tax policy endogeneity: Evidence from R&D tax credits. Econ. Innov. New Technol. 2018, 27, 809–833.

Czarnitzki, D.; Hanel, P.; Rosa, J.M. Evaluating the impact of R&D tax credits on innovation: A micro-econometric study on Canadian firms. Res. Policy 2011, 40, 217–229.

Dimos, C.; Pugh, G. The effectiveness of R&D subsidies: A meta-regression analysis of the evaluation literature. Res. Policy 2016, 45, 797–815.

Doh, S.; Kim, B. Government support for SME innovations in the regional industries: The case of government financial support program in South Korea. Res. Policy 2014, 43, 1557–1569.

Ernst, C.; Spengel, C. Taxation, R&D tax incentives and patent application in Europe. SSRN Electron. J. 2011.

Gramkow, C.; Angerkraavi, A. Could fiscal policies induce green innovation in developing countries? The case of Brazilian manufacturing sectors. Clim. Policy 2018, 18, 246–257.

Jose, M, Sharma, R. Effectiveness of fiscal incentives for innovation: Evidence from meta‐regression analysis. J Public Affairs. 2020;e2146. https://doi.org/10.1002/pa.2146

Klette, T.J.; Jarle, M. R&D investment responses to R&D Subsidies: A theoretical analysis and a micro-econometric study. World Rev. Sci. Technol. Sustain. Dev. 2012, 9, 169–203.

Lee, J.W. Government interventions and productivity growth in Korean manufacturing industries. J. Econ. Growth 1996, 1, 391–414.

Marino, M.; Lhuillery, S.; Parrotta, P.; Sala, D. Additionality or crowding-out? An overall evaluation of public R&D subsidy on private R&D expenditure. Res. Policy 2016, 45, 1715–1730.

Ning, L.; Li, J.C. Incentive effect of fiscal and taxation policies on enterprise technological innovation. Econ. Probl. 2019, 11, 38–45.

Peng, H.; Wang, G. Measurement and Analysis of the effects of Chinese government’s innovation subsidies. Res. Quant. Econ. Tech. Econ. 2018, 1, 77–93.

Wang, R.H.; Kesan, J.P. Do tax policies drive innovation by SMEs in China? J. Small Bus. Manag. 2020.

Yang, X.M.; Liu, W.L. Have fiscal R&D subsidies and tax incentives stimulated substantial innovation in manufacturing companies—Research based on propensity score matching and sample quantile regression. Ind. Econ. Rev. 2019, 10, 115–130.

Zhang, K.; Liu, X.L.; Fu, Z.R. Value added tax relief, enterprise tax burden, and innovation input: An analysis based on 2013-2015 survey Data. Commer. Res. 2017, 11, 39–45.

Zhang, N.; Du, J.T. The influence of fiscal and taxation policies on the innovation efficiency of high-tech enterprises—Based on the perspective of interaction. Tax Res. 2019, 12, 47–53.

Zúñiga-Vicente, J.Á.; Alonso-Borrego, C.; Forcadell, F.J.; Galán, J.I. assessing the effect of public subsidies on firm R&D investment: A survey. J. Econ. Surv. 2012, 28, 36–67.

Research hypotheses should be developed considering a wider literature framework. Please, provide a more effective literature support to hypotheses.

Please, add a section in which typologies of subsidies made available by the Chinese Government are illustrated and classified. Is there any difference between local and central government subsidies as in other countries?

Please, clarify the research design. As I understand, authors use 4 different models. Why? Is there any time effect? Time is a critical variable in innovation processes.

Although sample includes ICT companies, the subsectors have differences in terms of technology and market structure. To what extent such differences affect results?

Table 8, page 11: what do (1), (2), and (3) indicate? The same is for table 9. Please, make these tables more informative adding a further row.

Author Response

(The authors gave the same response as above.)

Round 2

Reviewer 1 Report

Dear Authors,

Good work!  I am satisfied with the improvements made in the paper.

Best regards

Reviewer

Author Response

Dear Editor and Reviewers,

Thank you very much for your consideration of our manuscript and request for a revised version.

Please find the attached manuscript entitled as “Impact of Chinese government subsidies on enterprise innovation- Perspective of the institutional environment, market structure, and corporate governance " to be reconsidered for publication at Sustainability.

We have copied and pasted all reviewers’ comments below, and addressed each one individually. As requested, we have listed the changes we have made including how and where. Please refer to the text marked as red for the listings of changes in the revised manuscript. As you will see, we have made every attempt to incorporate these suggestions as thoroughly as possible.

Yours sincerely,

Reviewer 3 Report

Authors have addressed some of my concerns. Particularly, they have improved the literature review section and hypotheses development. However, there are some further issues that need attention.

These issues are listed as follows.

Section 3.2.1. Model construction and variable selection: please, authors should explain why they use 4 equations, i.e. (1), (2), (3), (4), in the first part of this section.

Additionally, there is no time-dependency as these equations do not include a time variable. As I understand, time has been considered as a control variable. Why not considering time lag between subsidies and innovation? Is this a limitation of the study?

In Tables 8-9-10: even though notes clarify the meaning of (1), (2), ….I suggest to add “model specification” in the top raw of tables.

Author Response

(The authors gave the same response as above.)
